# Similarity between mutation spectra in hypermutated genomes of rubella virus and in SARS-CoV-2 genomes accumulated during the COVID-19 pandemic

Leszek J. Klimczak[1], Thomas A. Randall[1], Natalie Saini[2], Jian-Liang Li[1], Dmitry A. Gordenin[2]*

1 Integrative Bioinformatics Support Group, National Institute of Environmental Health Sciences, NIH, Durham, North Carolina, United State of America, 2 Mechanisms of Genome Dynamics Group, National Institute of Environmental Health Sciences, NIH, Durham, North Carolina, United State of America

* gordenin@niehs.nih.gov

## Abstract

Genomes of tens of thousands of SARS-CoV2 isolates have been sequenced across the world and the total number of changes (predominantly single base substitutions) in these isolates exceeds ten thousand. We compared the mutational spectrum in the new SARS-CoV-2 mutation dataset with the previously published mutation spectrum in hypermutated genomes of rubella—another positive single stranded (ss) RNA virus. Each of the rubella virus isolates arose by accumulation of hundreds of mutations during propagation in a single subject, while SARS-CoV-2 mutation spectrum represents a collection events in multiple virus isolates from individuals across the world. We found a clear similarity between the spectra of single base substitutions in rubella and in SARS-CoV-2, with C to U as well as A to G and U to C being the most prominent in plus strand genomic RNA of each virus. Of those, U to C changes universally showed preference for loops versus stems in predicted RNA secondary structure. Similarly, to what was previously reported for rubella virus, C to U changes showed enrichment in the uCn motif, which suggested a subclass of APOBEC cytidine deaminase being a source of these substitutions. We also found enrichment of several other trinucleotide-centered mutation motifs only in SARS-CoV-2—likely indicative of a mutation process characteristic to this virus. Altogether, the results of this analysis suggest that the mutation mechanisms that lead to hypermutation of the rubella vaccine virus in a rare pathological condition may also operate in the background of the SARS-CoV-2 viruses currently propagating in the human population.

## Introduction

RNA viruses can show a high mutation rate [1], which often results in fast emergence of viral quasispecies—populations of viruses differing in several genomic positions from the original virus [2]. Errors made by the RNA-dependent RNA-polymerase (RdRP) viral replicase are a

**Funding:** This work was supported by the US National Institute of Health Intramural Research Program Project Z1AES103266 to D.A.G.

**Competing interests:** The authors have declared that no competing interests exist.

source of mutations, however in coronaviruses some of these errors can be corrected by a proofreading RNA exonuclease ExoN [3, 4]. Another source of mutations comes from RNA base editing by two classes of endogenous enzymes: adenine deaminases (ADAR) and cytosine deaminases APOBECs (APOlipoprotein B mRNA Editing Complex like polypeptides) which have a broad range of functions spanning from site-specific editing of cellular mRNAs to inhibiting viral and retrotransposon proliferation [5–7].

ADARs (ADAR1 and ADAR2) are double-strand (ds) RNA-specific enzymes converting adenine to inosine (A to I). Since inosine pairs with cytosine, this will result in A to G changes after the next round of replication. The preference of ADARs for certain deamination motifs —reflecting a combination of immediate nucleotide context and the anticipated dsRNA formed by folding—was assessed for *in vitro* editing of several RNA substrates. Based on these data, software was developed aimed to assign predictive ADAR deamination scores to any A position in a given RNA molecule [8]. The ADAR editing sites that were deduced in RNAs of cultured stimulated immune cells [9] agreed with the preferences defined in the in vitro study. It remains to be established whether these preferences would hold for a wide variety of RNA substrates in conditions of controlled in vivo expression of either ADAR1 or ADAR2.

Unlike ADARs, the structure of APOBEC enzymes allow deamination only in single-strand (ss) RNA or in ssDNA. At least two APOBECs, APOBEC1 and APOBEC3A are capable of deaminating cytosine to uracil in RNA, however an RNA editing capacity of other APOBECs cannot be excluded [10–13]. Cytosine deamination in RNA creates the normal RNA base–uracil, which can be then accurately copied in subsequent rounds of RNA replication. DNA deamination motifs or mutation signatures (i.e., the immediate nucleotide contexts around deaminated Cs) of several human APOBECs were first defined in model microbial systems and then found in genomic DNAs of human cancers, where they can cause hypermutation clusters [14–19]. The preferred DNA deamination motif of APOBEC3G (A3G) is nCc to nTc (n = any base; the mutated nucleotide and the resulting nucleotide are capitalized). Other APOBECs show preference for the tCn to tTn deamination motif or to a more stringently defined trinucleotide. For example, both APOBEC3A (A3A) and APOBEC3B (A3B) prefer tCa as a target, both in the yeast model and in human cancers [15].

An indication of frequent RNA editing was recently found in the isolates of the hypermutated plus-strand ssRNA rubella vaccine virus from cutaneous granulomas of children with primary immunodeficiencies [20]. Altogether, genomes of six independent isolates of the hypermutated vaccine-derived viruses contained 993 mutations. Most changes were C to U in the genomic plus-strand RNA. These C to U changes showed high enrichment of a uCa to uUa RNA editing motif–a match to the characteristic A3A or A3B mutagenic motif in DNA. While the similarity between the C to U RNA editing motif in rubella virus and the DNA editing motifs strongly suggested the nature of the editing enzyme, signature motifs of APOBEC cytosine deamination in RNA are yet to be confirmed in a direct study involving expression of an APOBEC enzyme and collection of in vivo-editing spectrum data. The second most prevalent type of editing event was A to G change in the rubella plus or minus-strand, revealed as either U to C or G to A changes in the reported plus-strand sequence, respectively. These changes would be expected to result from ADAR editing. Minus-strand RNA in rubella virus as well as in Coronaviridae would often exist within completely- or partially double-stranded RNA [21, 22], which would be the right substrate for an ADAR. This strand is a template for the multiple rounds of transcription generating many plus-strand partial- or full-size genomic RNAs. Thus, an A to G editing event in the minus-strand of dsRNA at the beginning of replication cycle would be carried as a U to C change to multiple rubella virus genomes. A to G editing events in the plus-strand of the dsRNA intermediate may directly contribute to the mutation spectrum in plus-strand viral genomes or propagate the mutation via the subsequent

rounds of replication within the same cell. Besides ADAR editing, U to C (or complementary A to G) changes can result from uracil modifications by enzymes normally acting on specific uracils in tRNAs [23, 24]

In summary, the previous analysis of the mutation spectrum and mutational signatures of hypermutated rubella virus genomes provided a strong indication of hyperediting by APOBEC cytidine deaminases as well as suggested editing by ADAR adenine deaminases [20]. Both rubella and Coroniviridae are positive ssRNA viruses which produce many copies of the geno- mic positive RNA strand and also have dsRNA intermediates in their replication cycles [21, 22, 25], which can serve as substrates for APOBECs and for ADARs, respectively. Indeed, recent analyses suggested APOBEC and ADAR editing in SARS-CoV-2 based on an excess of C to U changes and A to G in sequencing reads from lavages of two COVID-19 patients or in genome alignments [26, 27]. Based on the similarity between the preferred RNA editing motifs in rubella virus and the APOBEC DNA hypermutation motifs, we sought to determine whether similar mutational signature motifs can be detected in a collection of 32,341 whole genome sequences of multiple SARS-CoV-2 isolates that have been sequenced during the current COVID-19 pandemic. We present here the evidence indicating a similarity between the RNA editing spectra and mutational signatures between the hypermutated rubella virus isolates and the load of editing changes accumulated in this collection of SARS-CoV-2 genomes. We also found several new trinucleotide-centered mutational motifs unique to SARS-CoV-2.

## Materials and methods

### SARS-CoV-2 genomes

SARS-CoV-2 genome sequences in the FASTA format were downloaded from https://www. epicov.org/epi3/frontend# at 13:10 EST on 2020/06/24 after applying the following download filters: (i) "complete sequence"; (ii) "high coverage"; (iii) "human"; (iv) "hCOV-19/. . .". The downloaded 32,341 FASTA entries were edited to remove spaces from FASTA headers (fatal defects for many tools) and reformatted to a consistent line length of 80 characters. Several samples with non-standard FASTA problems (many of them contain hyphens) that cannot be reasonably fixed and failed at the stage of alignment with the reference, therefore only 32,115 isolates were included into mutation calling.

### Mutation calls in SARS-CoV-2 genomes

Mutations in individual isolates were identified using MUMmer 3.23 ([28] and http:// mummer.sourceforge.net/) by making pairwise alignments with the original Wuhan isolate (GenBank entry NC_045512.2) using the command:

```
nucmer NC_045512.2.fasta query.fasta
```

The SNP variants output was generated using the command:

```
show-snps -T -Clr out.delta
```

and concatenating the individual results into a single tab-delimited text file.

For compatibility with other mutation analysis tools, the variant tables were created using the Mutation Annotation Format (MAF): https://software.broadinstitute.org/software/igv/ MutationAnnotationFormat but any suitable mutation representation format can be used instead. Functional annotation of the mutations was performed using the standard protocol of ANNOVAR ([29] and https://doc-openbio.readthedocs.io/projects/annovar/en/latest/) based on the genome annotations in GenBank entry NC_045512.2.

Out of 251,481 mutations initially called in 32,115 isolates, 251,273 were retained after removing redundant DNA symbols (anything but A,C,G,T) as well as mutation calls separated by less than 20 nt from either end of the reference, of which 243,454 were SNVs in 32070

isolates. Those mutations, redundantly spread in multiple isolates, were collapsed into a non-duplicated MAF designated as NoDups (up to three substitution types at each individual base position in the genome) of 13,736 mutations, 12,156 of which were SNVs. The NoDups filtered MAF was further subdivided into two MAFs: (i) NoDupsNonFunc MAF containing only 4,740 base substitutions that either caused a synonymous change in protein or were located in non-coding regions and therefore were annotated as non-functional; (ii) NoDupsFunc MAF containing only 7,416 base substitutions causing either aminoacid change or protein-truncation and therefore annotated as functional.

## Rubella virus genome and mutation data

The set of 993 base substitutions identified in six hypermutated isolates of rubella RA27/3 vaccine strain listed in MAF format were obtained from a previous study [20]. RA27/3 strain reference sequence GenBank entry FJ211588 was used for RNA-fold and nucleotide context annotations. Rubella mutation calls were compared with de-duplicated sets of SARS-CoV-2 mutation calls from 32,115 isolates contained in three versions of filtered MAFs (see "Design of the analysis" in Results).

## Comparison of SARS-CoV-2 and rubella virus base substitution spectra

The first indication of certain mutagenic mechanisms prevailing in generation of mutation load is a non-uniform distribution of base substitutions. Base substitution counts in each virus depend on both the relative probability of a given base substitution within the group of three possible substitutions of a given base and on the prevalence of each of four bases in a viral genome. Thus, in order to correct for the latter, we calculated densities of each of twelve possible base substitutions in each SARS-CoV-2 and rubella MAFs, dividing a base substitution count by the number of the mutated base in the reference sequence. We then assessed similarity of base substitution densities distributions between rubella virus and each of SARS-CoV-2 filtered MAF using non-parametric Spearman correlation with the null hypothesis that, there is no positive correlation between spectra in rubella and SARS-CoV-2.

## Statistical evaluation of mutagenesis in trinucleotide-centered mutation motifs

Calculating enrichment and statistical evaluation of mutagenesis in a small number of trinucleotide-centered mutation motifs identified from mechanistic knowledge turned productive in our prior assessments of mutagenesis associated with established mechanisms and known preference to certain trinucleotide motifs [15, 20, 30, 31]. In this study we extended statistical evaluation to all 192 possible trinucleotide centered motifs.

Trinucleotide and single-nucleotide frequencies in the genomic background were calculated using two alternative methods:

1. context-based–counts in the 41 nt windows centered around each mutation location;

2. reference-based–counts in the whole reference genome.

In both cases, Jellyfish ([32] and https://www.cbcb.umd.edu/software/jellyfish/) was used to calculate the counts of tri- and mononucleotides (k-mers with k equal 3 or 1, respectively) in the appropriate FASTA sequences (multiple FASTA entries for context, single entry for the reference). Each of the three substitution types in each of the 64 trinucleotides (total of 192) centered around the mutated base were counted with a set of 192 counters based on string-indexed arrays implemented as simple commands in Awk.

Counts of single nucleotide mutations, mutated trinucleotide motifs as well as trinucleotide and single-nucleotide frequencies in the genomic background were used to calculate enrichment with mutagenesis in each of 192 motifs over the presence expected for random mutagenesis as follows:

Enrichment (E) of xYz to xMz mutations calculated as

$$E(xYz \text{ to } xMz) = ((xYz \text{ to } xMz\_counts)/(Y \text{ to } M\_counts))/(xyz\_counts/y\_counts),$$

Where

Y and M are the original nucleotide and the nucleotide resulting from mutation, respectively,

y is the nucleotide in the context identical to Y in mutation motif

x and z are 5' and 3' flanking nucleotides in a motif, respectively

Statistical evaluation of Enrichment values was performed by two-tailed Fisher's exact test p-value comparing two ratios:

((xYz:M_counts)/(Y:M_counts-xYz:M_counts)) *vs* (xyz_counts/y_counts-xyz_counts)

P-values were then corrected for multiple hypotheses testing by Benjamini-Hochberg FDR including all 192 motifs. Only values passing FDR = 0.05 were considered statistically significant.

A minimum estimate of the number of mutations in a sample caused by xYz to xMz specific mutagenesis in excess of what would be expected by random mutagenesis was calculated as follows:

$$xYz : M\_MutLoad\_MinEstimate = [xYz : M\_counts] * [(xYz : M\_enrich - 1)/xYz : M\_enrich].$$

Calculated values were rounded to the nearest whole number. xYz:M_MutLoad_MinEstimate was calculated only for samples passing FDR = 0.05, signifying a statistical over-representation of motif-specific mutagenesis. Samples with FDR>0.05 received a value of 0.

## Statistical evaluation of preference to loop or stem locations in predicted RNA secondary structure

The RNAfold function of the ViennaRNA Package 2.0 [33] was used to determine the secondary structure of the complete FASTA sequences of the reference genomes for the SARS-CoV-2 virus (NC_045512.2) and the RA27/3 rubella vaccine virus (FJ211588). A sample command for generating the secondary structure of SARS-CoV-2 genome shown below:

```
RNAfold -d2—noLP < nc_045512.2.ref.fasta > nc_045512.2.ref.RNAfold.
out
```

The output for each analysis (.out) in dot-bracket notation was input into BBEdit (https://www.barebones.com/products/bbedit/) and all characters in both sequence and notation rows were made space delimited. Each of these rows were pasted into Excel and turned into space delimited cells. Sequence and notation were separately copied and pasted using the "Transform" function into a new Excel spreadsheet. A column with the nucleotide position was added and the file saved as a tab delimited text file *RNAfold.txt. For each resulting file the first column was the nucleotide position, the second column is the nucleotide, and the third column was the annotation of that nucleotide in dot-bracket notation. The *RNAfold.txt files were used to add a stem-loop annotation column "RNAfold" to all MAF files using the vlookup function in Excel and saved as a tab delimited text file.

For searching for motifs and trinucleotides, *RNAfold.txt files were used to create a searchable text files as follows. Columns two and three of each file were copied into a two new text files. On command line, the two columns in each file were merged using

```
awk '{$(NF-1) = $(NF-1)""$NF;$NF = ""}1' OFS = "\t"
```

The output file from this was opened in BBEdit, the line breaks were removed, resulting in a file containing nucleotides and annotation of those nucleotides in a single row as an interleaved and searchable format as CoV2_annot_final.txt and Rubella_annot_final.txt.

These files, displayed in BBEdit, were used to separately count all single nucleotides and all 64 trinucleotides classified as either stem or loop location based on the stem or loop annotation of the individual nucleotide position or of a central position in each trinucleotide.

Statistical evaluation of differences between loop vs stem single base substitution mutagenesis or trinucleotide motif associated mutagenesis was by comparing mutation densities in loop vs stem:

mutLoop/refLoop—density of a substitution type or a trinucleotide motif mutation type in loops

where

mutLoop and refLoop are counts in loops of a given type of events mutations or nucleotides in reference, respectively,

and

mutStem/refStem—density of a substitution type or a trinucleotide motif mutation type in stems

where

mutStem and refStem are counts in stems of a given type of events mutations or nucleotides in reference, respectively.

Statistical evaluation of loop vs stem mutagenesis was performed by two-tailed Fisher's exact test comparing ratios (mutLoop/(refLoop-mutLoop)) and (mutStem/(refStem-mutStem)) for either base substitutions or for trinucleotide motifs. Fisher's exact test p-value was corrected by Benjamini-Hochberg for the set of 12 possible base substitutions or for 16 possible tri-nucleotide centered around a given base substitution.

## Results

### Design of the analysis

The overarching hypothesis of this study was that some of the processes generating RNA mutation load in population of SARS-CoV-2 genomes are similar (but not necessarily identical) to the processes that generated changes in genomic RNAs of hypermutated rubella viruses. For that purpose, we obtained the viral genome FASTA files and processed them to obtain unique mutation calls and the mutation signatures as outlined in Fig 1.

32,341 FASTA files were downloaded from the GISAID Initiative [34] web site (https://www.gisaid.org/) on 06/22/2020, each containing a consensus whole genome-sequence of a SARS-CoV-2 virus isolated from a human subject and sequenced at high coverage. Based on the published analysis of the GISAID data for a subset of around 4000 of SARS-CoV-2 isolates across the world performed with the use of the Nextstrain package ([35] and https://nextstrain.org/ncov/global?l=clock), an average lineage of SARS-CoV-2 virus successfully transmitted from one subject to another would accumulate approximately 22 base substitutions per year (12–13 base substitutions for the period of December—June, 2020); a similar estimate was also obtained in [36]). The final FASTA sequence files of the individual isolates in GISAID represent a consensus derived from high coverage sequencing reads and contain information about the mutations present with high frequency in a sequenced viral isolate and therefore belong to

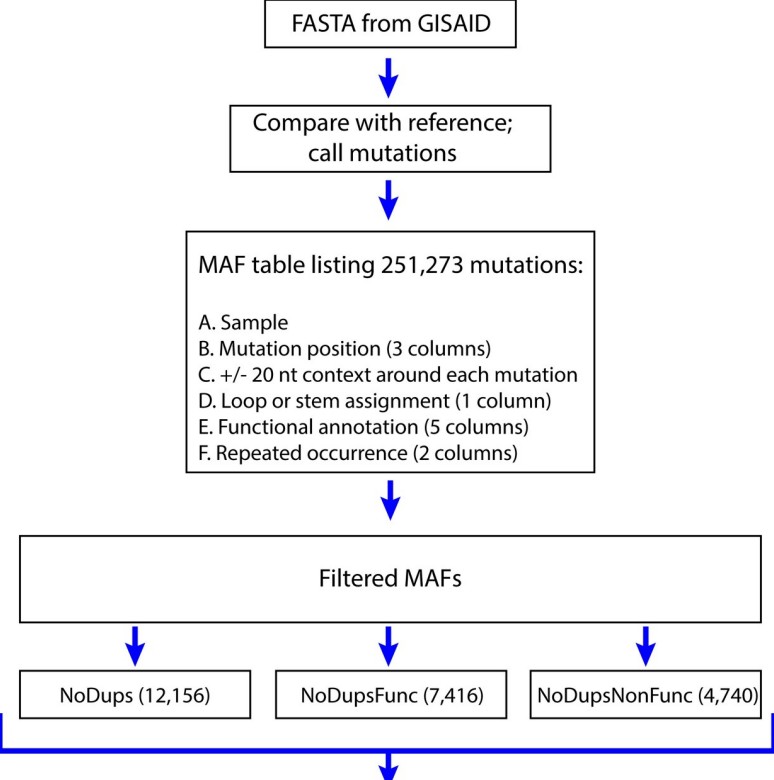

**Fig 1. Analysis workflow.** MAF–mutation annotation format table. Each filtered MAF combines mutations from all samples into a single dataset. Details of mutation call filtering and grouping as well as abbreviations are explained in the text of the "Design of the analysis" section.

viral particles capable of proliferation. We aligned each sequence against the sequence of presumably the earliest isolate of a SARS-CoV-2 genome (NC_045512. 2) and listed each change in a separate row of a mutation annotation file (MAF) Fig 1 and S1A Table. We annotated each of 251,273 mutation events in each isolate by surrounding +/- 20 nucleotides of genomic context around position of each mutation, by location in one or in several overlapping ORFs, by potential amino acid change or protein truncation effect, as well as by location of a change in self complementary area (predicted stem) or outside of such area (predicted loop) in plus-strand genomic RNA. Many independent isolates could have originated from the already mutated virus spreading the same mutation(s) into genomes of multiple (up to thousands) downstream isolates (see column "times_refPos_mutated_inMAF" in S1A Table). Therefore, we also annotated each mutation in a sample by the number of different samples in which such a mutation was found. Since each genome of an individual isolate contained only few mutations and many of these mutations were identical in multiple isolates, we built our analysis to evaluate the overall spectrum of non-redundant mutation events that have accumulated in the human population through the current pandemic rather than the mutation spectra in

individual isolates We de-duplicated the starting MAF and created three groups of mutations (S2A–S2C Table). The first group contained a pooled non-redundant set with no duplicates (NoDups) that listed each individual mutation only once regardless of how many isolates contained the same mutation (total 12,156 mutation events). While the individual isolates are not listed in this group any more, it contains a set of distinct events most closely representing the spectrum of unrelated mutation events rather than a complex downstream process of distributing mutant forms in human population. However, even in the NoDups list (S2C Table) the base substitutions in many positions could be under positive or negative selection, which could skew the spectrum of the observed changes from the mechanistic mutation spectrum that accurately reflects the underlying mutagenic processes. Therefore, we subdivided this group based on whether the changes yielded functional effects in the SARS-CoV-2 genome. Non-synonymous amino acid changes and changes introducing or removing stop codons were designated as functional (Func), while synonymous changes or changes outside ORFs were designated and non-functional (NonFunc). The content of NoDupsNonFunc group (S2A Table) would be the least affected by functional selection and thus, most accurately represent the impact of unconstrained mutational processes operating on the viral genome. While this group is smaller, it still contains a sufficient number of changes (4,740 mutation events) for detecting trends in the mutational patterns. The mutation spectrum of 7,416 NoDupsFunc events (S2B Table) was also analyzed. Each of the three SARS-CoV-2 mutation spectra was compared to the combined mutation spectrum (993 base substitutions) from six independent isolates originated from the hypermutated rubella-vaccine virus [20] (S3A Table). Unlike many SARS-CoV-2 isolates, where individual mutated event could be carried from one isolate to another, each rubella virus isolate contained mutations that had occurred independently from the vaccine virus in each subject. Thus, the total of the mutational events in six rubella isolates was, at least in part, representative of the mutation spectrum. However, mutation spectra in each rubella isolate may represent an unknown level of selection. Indeed, a number of mutations was observed in more than one rubella virus isolate (see column "times_refPos_mutated_inMAF" in S3A Table). Some level of selection was also indicated by analysis of synonymous and nonsynonymous substitutions in each codon [20]. Therefore, we made separate comparisons of the rubella virus mutation spectra with each of the three SARS-CoV-2 non-redundant MAFs (Fig 1): the non-duplicated mutation events (NoDups), and its two subsets– the non-duplicated mutation events with potential of functional significance (NoDupsFunc) and the non-duplicated non-functional mutation events (NoDupsNonFunc).

All mutations are reported based on the plus (genomic) strand of the virus. We started from conventional comparisons of all possible single base substitutions and the mutation preference for potential loop or stem parts of ssRNA secondary structure. Unlike in our previous analysis of the mutation spectrum and signatures in the genomes of hypermutated rubella virus isolates, where we followed only a limited set of motifs based on specific hypotheses, we used here an "agnostic" approach analyzing all 192 possible trinucleotide-centered mutation motifs for enrichment in the viral genomes. We also used existing software to calculate ADAR editing scores [8]. Overall, our methodology allows to detect the mutational signatures that predominate in the viral genomes. Comparisons with the hypermutated rubella genomes further demonstrated the similarities in the mutational processes operating on both viral genomes.

## Similarity of base substitution spectra between hypermutated rubella virus genomes and SARS-CoV-2

We compared the distribution of densities of the 12 possible single base substitutions (counts of each base substitution normalized by the presence of the unmutated base in the genome).

While density distributions reflect the contributions of different mutagenic processes in each dataset, density values for specific base substitutions cannot be compared directly between two viruses because they were obtained from vastly different genome numbers. Importantly, there was a statistically significant similarity between the distributions of base substitution densities in rubella and in each of three filtered SARS-CoV-2 MAFs as well as a similarity in several prevailing types of base substitutions (Fig 2 and S4 Table).

In both viruses, there was a very high frequency of the C to U changes, consistent with the hypothesis of cytidine deamination in the plus-strand (genomic) RNA. C to U changes in the minus-strand, which would be reported as G to A in the plus-strand, were less abundant in both viruses. Another class of highly abundant changes in both viruses were U to C changes in the plus RNA strand which could originate from A to G changes caused by ADAR adenine deaminase in the minus-strand. The corresponding A to G changes in the plus-strand were less abundant in rubella but were comparable with the C to U changes in SARS-CoV-2.

A prior study of hypermutated rubella genomes found small, but statistically significant increase in ADAR scores (calculated as described by [8]) in U to C and A to G ADAR-like base substitutions compared to two other types of substitutions in U or A nucleotides [20]. However, no statistically significant increase in ADAR scores was found for the U to C and A to G changes in the SARS-CoV-2 dataset analyzed in a similar way (S1 Fig and S1 Data). Since the ADAR score tool was developed based on in vitro deamination of a perfectly paired dsRNA substrate, there could be a difference in sequence preferences between this substrate and the actual substrate of in vivo editing of SARS-CoV-2 RNA. Alternatively, abundant U to C and A to G ADAR-like editing could be due to mechanisms not involving ADARs.

The only apparent discrepancy between the two viruses was in a high density of the G to U changes in the plus-strand of SARS-CoV-2, while they were nearly absent in rubella. We note that the density of G to U changes in minus-strand (reported as the complementary C to A changes in plus-strand in Fig 2) was similar to other low abundant changes in both viruses. A possible origin of the increased G to U changes in SARS-CoV-2 genomes will be detailed in Discussion.

### Several types of base substitutions show preference for regions prone to loop formation in viral RNA secondary structure

A high abundance of C to U (or G to A) mutations was already noticed in several recent analyses of SARS-CoV-2 mutation data and inferred to either APOBEC mutagenesis or to errors in RdRp copying of the minus-strand [26, 36, 37]. C to U mutations in RNA can be also caused by non-enzymatic deamination of cytidines similar to such deamination described in DNA [16, 38]. Recently it was revealed that APOBEC3A has a preference for deaminating cytosines in regions prone to forming loops in ssDNA secondary structure [39]. Therefore we annotated all positions in the SARS-CoV-2 and rubella genomes for either preference for loop or stem location in potential secondary structure formed by the RNA plus-strand ([33] S1B and S3B Tables and Methods). We then compared mutation counts in loop vs stem for each type of base substitutions (Fig 3 and S5 Table).

In both viruses there was a highly significant preference for loop location with C to U changes in plus-strand. The second type of base changes prevalent in both SARS-CoV-2 and in rubella, the U to C changes in plus-strand (corresponding to A to G changes in minus-strand) did not show statistically significant differences between loop and stem. If the U to C (A to G) changes were to come from ADAR adenine deaminase acting on dsRNA, secondary structure effects of ssRNA intermediate folding would not be expected. Alternatively, these changes could be not ADAR driven. The only other type of changes showing statistically

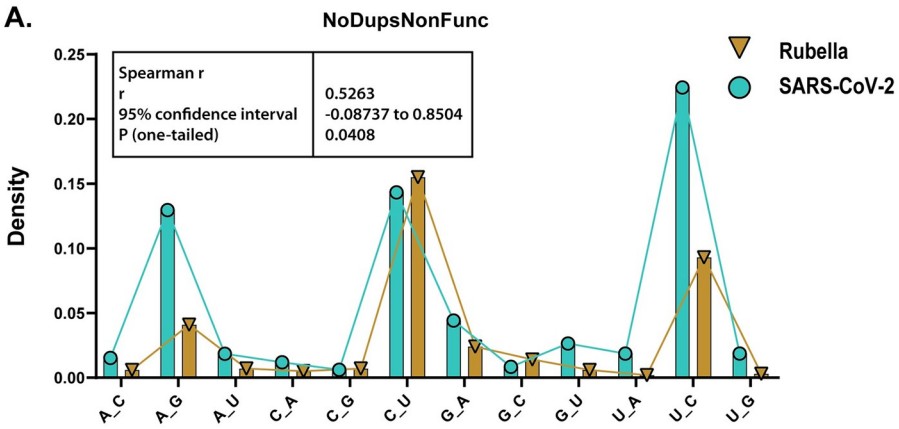

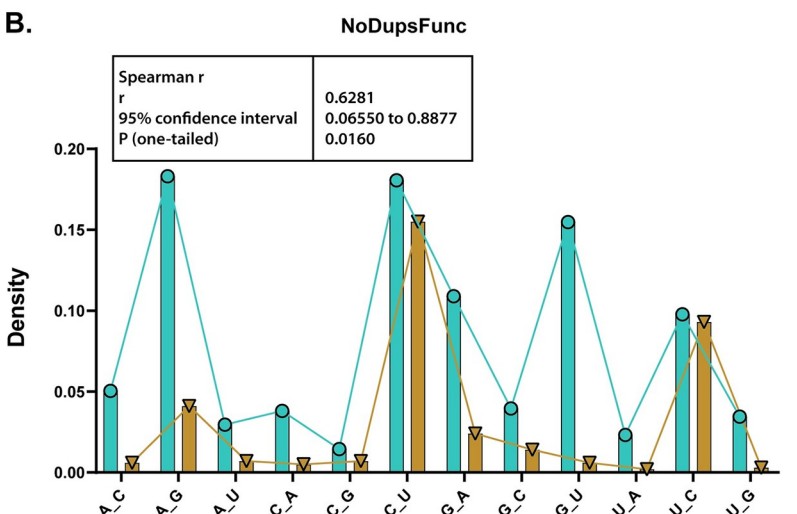

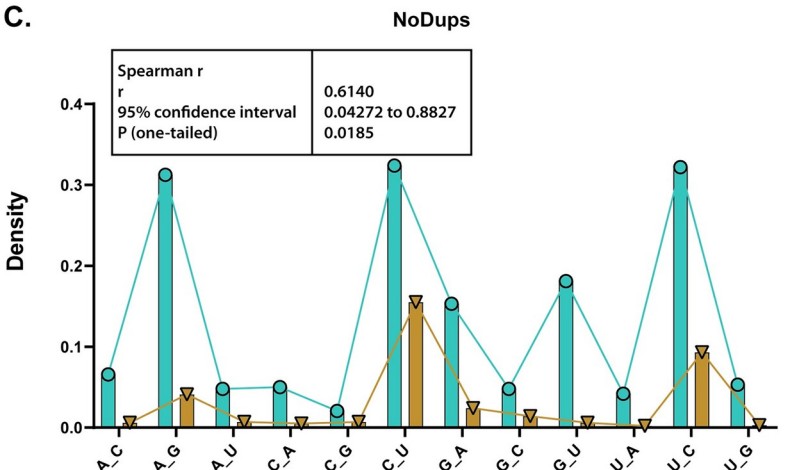

**Fig 2. Comparison of base substitution spectra between rubella virus and SARS-CoV-2 datasets from filtered MAFs.** Creation of filtered SARS-CoV-2 MAFs and their abbreviated names NoDupsNonFunc, NoDupsFunc, NoDups, are described in Fig 1 and in associated text. The spectrum from each SARS-CoV-2 filtered MAF was compared with rubella mutation spectrum. Bars represent densities of base substitutions in each dataset calculated by dividing counts of each base substitution by counts of the substituted base in the reference sequence. Connecting lines

visualize overall parallelism between rubella and each filtered MAF. Insert boxes show Spearman r, its 95% CI, and one-tailed p-value for hypothesis about positive correlation between rubella and SARS-CoV-2 spectra. Source data are in S4 Table.

significant difference between loop and stem locations in all groups of SARS-CoV-2 mutations were G to U changes. Same as for the base substitution spectra, this outstanding feature of the G to U changes showed up in SARS-CoV-2, but not in rubella (see above).

## Mutational motif preferences in SARS-CoV-2 and rubella virus genomes suggest APOBEC cytidine deaminases as a source of C to U base substitutions in the plus RNA strand

Since base substitution spectra in SARS-CoV-2 and rubella virus are correlated, it is likely that they also shared common mechanisms which generated these changes. It is well established that several mechanisms of mutagenesis in DNA of tumors and normal cells can have not only distinctive base substitutions spectra but also diagnostic preference for trinucleotide mutation motifs [30, 40, 41]. Currently there is very little information about motif preference in RNA editing or mutagenesis. Therefore, we assessed enrichment using all possible 192 trinucleotide mutation motifs (96 in plus and 96 in minus RNA strand) of each virus. Enrichment values for each motif were calculated based on counts of mutations in a motif normalized for the motif content in the genomic background (see Methods). Statistical evaluation of enrichments showed significance for several motifs even after FDR<0.05 correction to individual P-values was applied (S6A–S6C Table). However, base substitutions for the most-enriched motifs were present in low numbers, so these results require validation in independent studies (also see Discussion). Therefore, we concentrated on the motifs representing the most abundant types of base substitutions present in both viruses, i.e., on the C to U and their complement G to A, as well as U to C and their complement A to G changes. For statistically-significant enriched trinucleotide motifs containing one of these four base substitutions, we calculated the minimum estimates of mutation load (MutLoad) that can be assigned to mechanism(s) with preference for a significantly enriched motif (Fig 4, S6A–S6D Table and Materials and methods).

The only revealed similarity between statistically-significant enriched motifs in rubella virus and in SARS-CoV-2 was for the uCn to uUn changes, consistent with the tCn to tTn ssDNA mutagenesis specificity of a subgroup of APOBEC cytidine deaminases. However, even within the APOBEC-like group of motifs there was a difference between strong enrichment with uCa to uUa motif in rubella and the lack of statistically significant preference for this motif in SARS-CoV-2. There were also three groups of motifs significantly enriched in SARS-CoV-2, but not in rubella (see Table 1 and Discussion for possible mechanistic assignment of these motifs).

We also assessed the potential loop vs stem preference for trinucleotide motifs containing C to U and G to U single base substitutions that showed overall loop vs stem preference. None of trinucleotide motifs containing C to U base substitutions showed loop or stem preference in selection-free SARS-CoV-2 NoDupsNonFunc filtered dataset and in rubella (S2 Fig and S7A and S7D Table). Several C to U containing trinucleotide motifs in SARS-CoV-2 datasets, where functional selection cannot be excluded, showed statistically significant bias towards mutations in loops (S2 Fig and S7B and S7C Table), however more data accumulation is required in order to exclude the confounding effects of functional selection in specific sites. No loop vs stem preference was detected in trinucleotide motifs containing G to U substitutions either in SARS-CoV-2 or in rubella (S3 Fig and S8A–S8C Table).

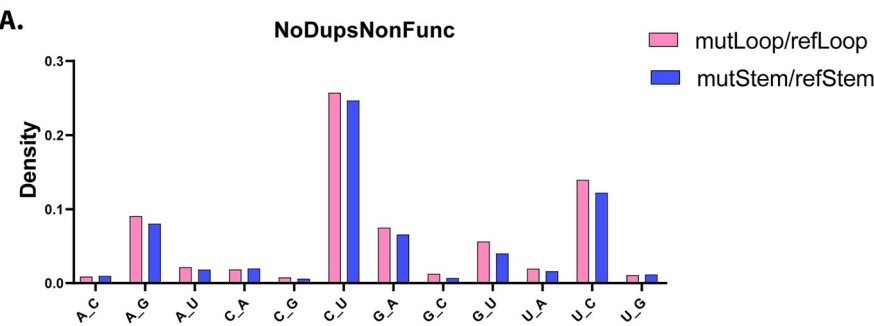

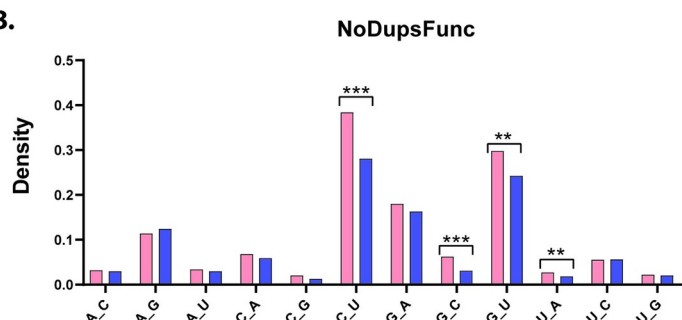

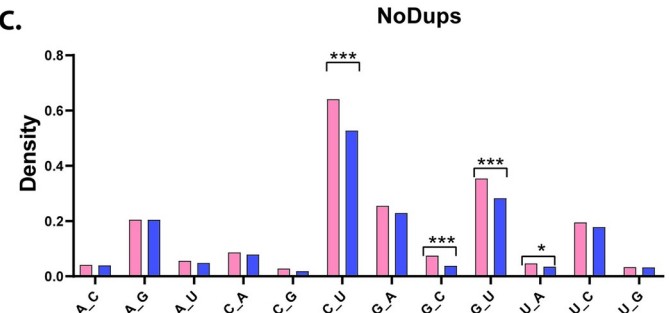

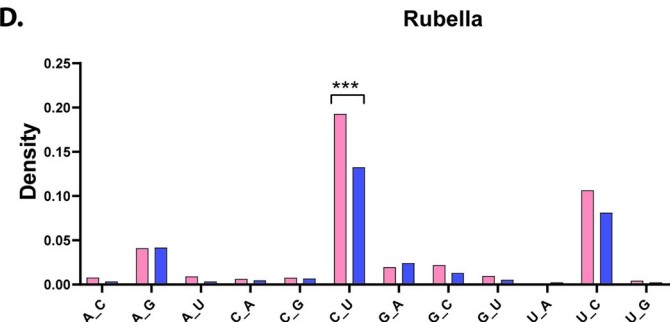

**Fig 3. Comparison of base substitution mutagenesis between locations prone to loop or stem formation in viral RNA genomes.** Creation of filtered SARS-CoV-2 MAFs and their abbreviated names NoDupsNonFunc, NoDupsFunc, NoDups, are described in Fig 1 and in associated text. Bars represent densities of base substitutions in stem- or in loop-forming sections. Densities are calculated by dividing counts of each base substitution in either loop or in stem by counts of the substituted base in the loop-forming or in stem-forming regions of the reference sequence. Statistical

comparison between mutagenesis in stem vs loop for every base substitution was done by two-tailed Fisher's exact test. P-values were considered after correcting by FDR. Brackets indicate pairs passing FDR = 0.05. *<0.05, ** <0.005, *** <0.0005. Source data including exact p-values are in S5 Table.

In summary, our agnostic analysis of trinucleotide signature motifs demonstrated that the uCn (tCn) APOBEC-like mutagenesis, which is a major component in rubella virus hypermutation, also contributes towards the mutations accumulated in the genomes of infectious SARS-CoV-2 spreading in the current pandemic.

## Discussion

Previously, we demonstrated that hypermutation of the live attenuated rubella vaccine virus [42] can generate infectious virus particles in immunocompromised children [20]. Based on this work, we hypothesized that similar mutagenic processes may act upon the genomes of

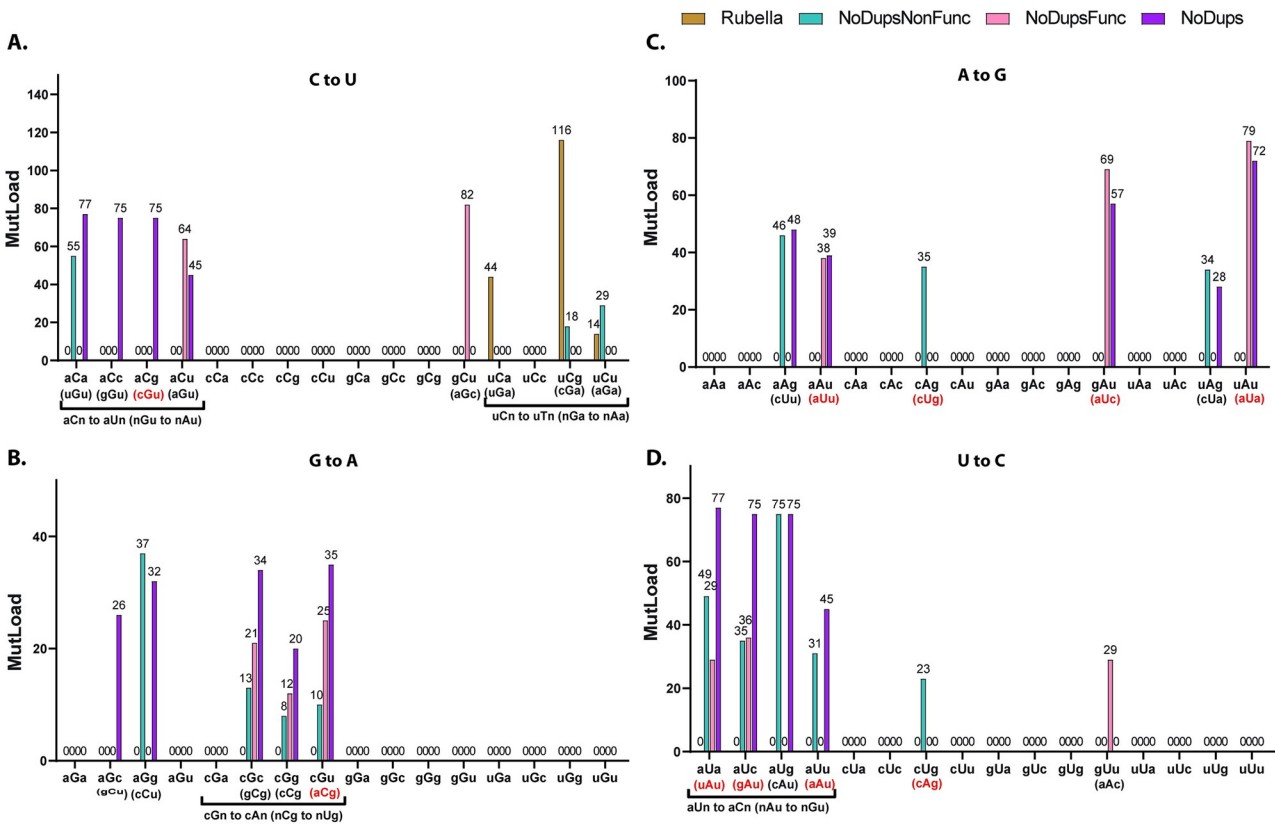

**Fig 4. Trinucleotide-centered mutation motifs with statistically significant enrichment over random mutagenesis.** Creation of filtered SARS-CoV-2 MAFs and their abbreviated names NoDupsNonFunc, NoDupsFunc, NoDups are described in Fig 1 and in the associated text. The order of data in each group is the same as in the panel's legend—Rubella, NoDupsNonFunc, NoDupsFunc, NoDups. Zero values are shown for the convenience of following the order of values within each group. Bars represent minimum estimates of mutation load that can be assigned to motif-specific mutagenic mechanism (MutLoad) as described in Methods. Statistical evaluation of enrichments was done by two-tailed Fisher's exact test and corrected by FDR including p-values for all 192 possible trinucleotide-centered base substitution motifs. MutLoad for FDR>0.05 = 0. Only results for motifs which included the most frequent base substitutions in the plus-strand, C to U, G to A, A to G, U to C, are shown. Reverse complement motifs in the plus-strand corresponding to the statistically significant motifs mutated in the minus-strand are shown in parentheses. If both plus-strand motifs in the reverse complement pair were statistically-significantly enriched in at least one dataset (in rubella or in a filtered SARS-CoV-2 MAF), they are highlighted in red font. Source data including calculations for all 192 motifs are in S6 Table.

**Table 1. Analyses of base substitutions prevailing in rubella virus and in SARS-CoV-2 plus-strand genomic RNAs.**

| Feature | Virus | C to U | G to A | A to G | U to C |
|---|---|---|---|---|---|
| Prevalence of a base substitution | rubella | High frequency | Frequent; Less frequent than C to U | High frequency; Less frequent than U to C | High frequency; More frequent than A to G |
|  | SARS-CoV-2 | High frequency | Frequent; Less frequent than C to U | High frequency | High frequency |
| Secondary structure element preferred by base substitution | rubella | Prefers loops over stems | ND | ND | ND |
|  | SARS-CoV-2 | Prefers loops over stems | ND | ND | ND |
| Enriched trinucleotide motif(s) | rubella | uCn to uUn | ND | ND | ND |
|  | SARS-CoV-2 | uCn to uUn; aCn to aUn | cGn to cAn (reverse complement for nCg to nTg in minus-strand); | nAu to nGu (reverse complement to motif preferred by U to C) | aUn to aCn (reverse complement to motif preferred by A to G) |
| Suggested mechanism | rubella and/or SARS-CoV-2 | (i) Frequent C to U editing by tCn (uCn) -specific APOBEC(s) in plus-strand; (ii) new motif aCn to aUn | (i) Increased C-deamination in nCg (CpG) minus-strand motif (SARS-CoV-2 only); (ii) No statistical support to APOBEC editing in minus-strands of either virus | A to G editing by ADAR(s) in plus-strand | A to G editing by ADAR(s) in minus-strand |

ND–not detected

other similar plus-strand RNA viruses like SARS-CoV-2. The large-scale sequencing efforts producing genomes of tens of thousands of SARS-CoV-2 isolates allowed us to accurately identify mutations, build a mutation catalog for this virus, highlight similarities with hypermutated rubella and reveal unique features of SARS-CoV-2 mutagenesis (summarized in Table 1).

In agreement with the previous analyses performed separately for rubella [20] and for SARS-CoV2 [26, 27, 36, 37] comparisons between SARS-CoV-2 and hypermutated rubella strains demonstrated that the base substitution spectra correlate between the two viruses. Two types of base substitutions–C to U (and its complementary G to A) and A to G (and its complementary U to C), expected from endogenous mutagenesis by APOBECs and ADARs, respectively, prevailed in both viruses (Fig 2).

Since RNA can form secondary structures, any mutagenic processes active upon ssRNA would preferentially be formed in the loop regions of the secondary structures. Analysis of A to G and U to C substitutions consistent with the biochemical specificity of ADARs did not reveal any preference for mutagenesis in loops versus stem regions. ADARs are known to act on dsRNA substrates. Thus, if ADARs did in fact contribute to induction of these substitutions, they should be acting on a dsRNA form, wherein we do not expect RNA to fold into secondary structures. We also found in SARS-CoV-2, but not in rubella, statistically significant enrichment of the nAu to nGu mutations along with its reverse complement aUn to aCn. While these could reflect adenine deamination by one of the ADARs in either strand of a dsRNA intermediate, these motifs are different from the motif preference revealed by *in vitro* editing of artificial dsRNA substrate [8]. Also, there was no increase in ADAR scores in the SARS-CoV-2 A to G or U to C mutations (S1 Fig and S1 Data), thus indicating that ADARs may not be the primary source of these changes. If A to G or U to C changes are really stemming from ADAR activity, it is possible that these enriched motifs are preferred by ADARs in SARS-CoV-2 but not in RNA substrates used to generate the editing consensus in [8]. Alternatively, A to G and U to C changes could be caused by one or more mechanisms unconnected to ADARs.

Unlike A to G changes, in-depth analysis of C to U substitutions revealed that they were predominantly present in the RNA plus-strands of both viruses and demonstrated a preference for loops versus stems in the RNA secondary structure (Fig 3 and S5 Table). This phenomenon

is similar to the preference of ssDNA and mRNA editing in loops by the APOBEC3A cytidine deaminase [39, 43], which so far is the only RNA-editing APOBEC explored for this feature. Agnostic analysis of enrichments in all 192 possible trinucleotide mutation motifs highlighted statistically significant excess of uCa to uUa motif in rubella, however these changes were not prevalent in SARS-CoV-2. Mutations in plus-strands of both viruses showed statistically significant enrichments with uCg to uUg and uCu to uUu motifs (Fig 4A and 4B). These motifs belong to a group uCn to uUn (tCn to tTn in DNA) which is characteristic of several APOBEC cytidine deaminases ([15] and references therein). We note that although these signatures were enriched in the non-functional mutations (NoDupsNonFunc), they did not pass the 0.05 FDR threshold in filtered datasets that included mutations with potential functional effects (NoDupsFunc). These differences in the mutation signatures between SARS-CoV-2 and rubella may be due to different APOBEC family members performing editing or due to the confounding presence of other sources of C to U mutagenesis, such as spontaneous cytosine deamination that frequently occurs in ssDNA [44] or oxidative mutagenesis capable of generating C to T mutations in ssDNA *in vivo* [45, 46]. In support of the role of oxidative damage in SARS-CoV-2 genomes, is the increased prevalence of G to U substitutions which is consistent with the oxidation of guanines in the RNA plus-strand (Fig 2). G to U changes could be caused by an increased level of oxidative damage generating 8-oxoG in viral RNA within cells or during sequencing library preparation [47, 48]. Frequent copying of 8-oxoG with A, would show up as G to U changes in the strand, where 8-oxoG was present. However, since we analyze the consensus sequences of the viral genomes and not individual reads, errors during library preparation would most likely be filtered out and would not be represented in the viral genome sequence. We also note that the recent study [27] indicated the overall low chance of sequencing errors reflected in SARS-CoV-2 consensus sequences in the dataset analyzed in that work. On the other hand, G to U changes were present only at low density in hypermutated rubella genomes indicating physiological differences between the two viruses.

There were two more groups of trinucleotide mutation motifs involving C to U (and complementary G to A) substitutions in plus RNA strand specifically enriched for SARS-CoV-2 (Fig 4A and 4B). The aCn to aUn (reverse complement nGu to nAu) group of motifs may represent a preference previously unknown for APOBECs in RNA or just a mutagenic mechanism yet to be defined. The cGn to cAn group of motifs seen in the plus-strand may be in fact due to mutations of the reverse complement motif nCg to nUg in the minus-strand. nCg to nTg (CpG to TpG) germline and somatic mutagenesis is universally present in DNA of species with 5-methylcytosine and is generated by systems specialized to mutagenesis in methylated CpG sequences. However various studies have demonstrated that CpG to TpG mutagenesis can occur independent of cytosine methylation [49, 50]. Several studies have shown that CpG dinucleotides are depleted in the genomes of SARS viruses indicating functional selection and/ or increased frequency of cytosine deamination in these viral genomes [51–54]. Our study shows with high statistical confidence that nCg to nUg (CpG to UpG) mutagenesis in the minus strand is enriched (Fig 4B) supporting the role of nCg- (CpG)-specific cytosine deamination in minus RNA strand in SARS-CoV-2 genomic mutagenesis.

In summary, comparison of base substitution spectra and signatures between hypermutated rubella virus isolates and the SARS-CoV-2 multi-genome dataset demonstrates both similarities and differences in the mutational processes active upon the two plus-strand RNA viruses. It is important to understand the mechanisms that contribute to mutagenesis of viral genomes, since hypermutation of even inactivated rubella vaccine virus was shown to generate reactivated viral particles [20]. We demonstrate here that the APOBEC-specific uCa to uUa changes that are highly enriched in hypermutated rubella, are much less prevalent in SARS-CoV-2. We propose that assessment of uCa to uUa signature in viral genomes can provide insights into

the potential hypermutation risk of SARS-CoV-2. Moreover, understanding the genomic mutational patterns is important for predicting virus evolution. Our study has highlighted several distinct features of SARS-CoV-2 mutational spectrum that, after validation with independent dataset(s) can be used to build predictive models for this and related SARS viruses.

## Supporting information

**S1 Fig. Mean values of ADAR scores.** ADAR scores were calculated using the Web tool http://hci-bio-app.hci.utah.edu:8081/Bass/InosinePredict. Source data are in S1 Data.
(PDF)

**S2 Fig. Comparison of trinucleotide-centered motif C to U mutation densities between locations prone to loop or to stem formation in viral RNA genomes.** Bars represent densities of base substitutions in stem- or in loop-forming regions. Densities are calculated by dividing counts of each motif mutations in either loop or in stem by counts of this motif in the loop-forming or in stem-forming regions of the reference sequence. Statistical comparison between mutagenesis in stem vs loop for every base substitution was done by two-tailed Fisher's exact test. P-values were corrected by FDR including 16 motifs containing C to U base substitution. Brackets indicate pairs passing FDR = 0.05. * <0.05, ** <0.005. Source data are in S7 Table.
(PDF)

**S3 Fig. Comparison of trinucleotide-centered motif G to U mutation densities between locations prone to loop or to stem formation in viral RNA genomes.** Bars represent densities of base substitutions in stem- or in loop-forming regions. Densities are calculated by dividing counts of each motif mutations in either loop or in stem by counts of this motif in the loop-forming or in stem-forming regions of the reference sequence. Statistical comparison between mutagenesis in stem vs loop for every base substitution was done by two-tailed Fisher's exact test. P-values were corrected by FDR including 16 motifs containing G to U base substitution. Brackets indicate pairs passing FDR = 0.05. * <0.05. Source data are in S8 Table.
(PDF)

**S1 Data. ADAR scores and complete ADAR analysis for SARS-CoV-2 filtered MAFs (contains source data for S1 Fig).**
(ZIP)

**S1 Table. Mutation calls and RNA fold prediction in SARS-CoV-2 genomes.** S1A. Complete list of mutation calls in all SARS-CoV-2 genomes in TCGA compatible Mutation Annotation Format (MAF); nucleotides named as in DNA. S1B. Annotation of predicted RNA-fold in SARS-CoV-2 reference positions.
(XLSB)

**S2 Table. SARS-CoV-2 filtered Mutation Annotation Files (MAFs).** S2A. NoDupsNonFunc —de-duplicated set of mutations from all samples of the dataset; non-functional. S2B. NoDupsFunc—de-duplicated set of mutations from all samples of the dataset; aminoacid changes or protein-truncating. S2C. NoDups—de-duplicated set of mutations from all samples of the dataset.
(XLSX)

**S3 Table. Mutation calls and RNA fold prediction in rubella virus genomes.** S3A. The list of 993 mutations in six rubella isolates (from [20]). Sequences are shown in DNA format

(T instead of U) to maintain compatibility with other outputs of the mutation signature R-script. S1B. Annotation of predicted RNA fold in rubella reference.
(XLSX)

**S4 Table. Counts and densities of single base substitutions in SARS-CoV-2 and in rubella virus.** S4A. Counts of base substitutions. S4B. Densities of base substitutions (Source data for Fig 2).
(XLSX)

**S5 Table. Comparison of base substitution densities between locations prone to loop or stem formation in viral RNA genomes.** (Source data for Fig 3). S5A. SARS-CoV-2, NoDups-NonFunc—de-duplicated set of mutations from all samples of the dataset; non-functional. S5B. SARS-CoV-2, NoDupsFunc—de-duplicated set of mutations from all samples of the dataset; aminoacid changes or protein-truncating. S5C. SARS-CoV-2, NoDups—de-duplicated set of mutations from all samples of the dataset. S5D. Rubella, all mutations.
(XLSX)

**S6 Table. Statistical evaluation of mutagenesis in 192 trinucleotide-centered mutation motifs.** Sequences are shown in DNA format (T instead of U) to maintain compatibility with other outputs of the mutation signature R-script. S6A. SARS-CoV-2, NoDupsNonFunc—de-duplicated set of mutations from all samples of the dataset; non-functional. S6B. SARS-CoV-2, NoDupsFunc—de-duplicated set of mutations from all samples of the dataset; aminoacid changes or protein-truncating. S6C. SARS-CoV-2, NoDups—de-duplicated set of mutations from all samples of the dataset. S6D. Rubella, all mutations.
(XLSX)

**S7 Table. Comparison of C to U trinucleotide motif substitution densities between locations prone to loop or stem formation in viral RNA genomes.** S7A. SARS-CoV-2, NoDups-NonFunc—de-duplicated set of mutations from all samples of the dataset; non-functional. S7B. SARS-CoV-2, NoDupsFunc—de-duplicated set of mutations from all samples of the dataset; aminoacid changes or protein-truncating. S7C. SARS-CoV-2, NoDups—de-duplicated set of mutations from all samples of the dataset. S7D. Rubella, all mutations.
(XLSX)

**S8 Table. Comparison of G to U trinucleotide motif substitution densities between locations prone to loop or stem formation in viral RNA genomes.** S8A. SARS-CoV-2, NoDups-NonFunc—de-duplicated set of mutations from all samples of the dataset; non-functional. S8B. SARS-CoV-2, NoDupsFunc—de-duplicated set of mutations from all samples of the dataset; aminoacid changes or protein-truncating. S8C. SARS-CoV-2, NoDups—de-duplicated set of mutations from all samples of the dataset. S8D. Rubella, all mutations.
(XLSX)

**S9 Table. Acknowledgments to research groups and individuals provided SARS-CoV-2 genome sequences to GISAID's EpiCoV™ Database.**
(XLSX)

## Acknowledgments

We thank Dr. Natalya Degtyareva and Mr. Adam Burkholder for critical reading of the manuscript, Mr. Michael Doubintchik for help in the initial evaluation of the on-line databases of SARS-CoV-2 genomes and Dr. Ludmila Perelygina for advice on revision. We gratefully acknowledge the authors, as well as the originating laboratories for submitting the sequences

to GISAID's EpiCoV™ Database on which this research is based; complete list of contributors is provided in S9 Table.

## Author Contributions

**Conceptualization:** Leszek J. Klimczak, Dmitry A. Gordenin.

**Data curation:** Leszek J. Klimczak, Thomas A. Randall, Natalie Saini, Dmitry A. Gordenin.

**Formal analysis:** Leszek J. Klimczak, Thomas A. Randall, Natalie Saini, Dmitry A. Gordenin.

**Funding acquisition:** Jian-Liang Li, Dmitry A. Gordenin.

**Investigation:** Natalie Saini, Dmitry A. Gordenin.

**Methodology:** Leszek J. Klimczak, Thomas A. Randall, Dmitry A. Gordenin.

**Project administration:** Jian-Liang Li, Dmitry A. Gordenin.

**Resources:** Jian-Liang Li.

**Software:** Leszek J. Klimczak, Thomas A. Randall, Jian-Liang Li.

**Supervision:** Dmitry A. Gordenin.

**Visualization:** Natalie Saini, Dmitry A. Gordenin.

**Writing – original draft:** Dmitry A. Gordenin.

**Writing – review & editing:** Leszek J. Klimczak, Thomas A. Randall, Natalie Saini, Jian-Liang Li, Dmitry A. Gordenin.

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
