## [Decision Letter · Decision Letter 0]

9 Sep 2020

PONE-D-20-23843

Similarity between mutation spectra in hypermutated genomes of rubella virus and in SARS-CoV-2 genomes accumulated during the COVID-19 pandemic

PLOS ONE

Dear Dr. Gordenin,

Thank you for submitting your manuscript to PLOS ONE. After careful consideration, we feel that it has merit but does not fully meet PLOS ONE’s publication criteria as it currently stands. Therefore, we invite you to submit a revised version of the manuscript that addresses the points raised during the review process.

1) Make the suggested minor changes requested by the reviewers.  While the additional data analysis suggested by Reviewer #1 might strengthen the manuscript, it is not essential for acceptance for publication (in case it is challenging or technically difficult).    

2) I want to emphasize that novelty is not a central criterion for publication in PLoS ONE - hence I would like to encourage you to not put too much focus on what is novel (compared to previously published competing studies) but also present and discuss comprehensively aspects where there is overlap (as the tools employed differ between the studies).  

3) Please contact me about anything that is unclear in the reviewers comments (or if there are difficulties in making the suggested changes), so that a I can help to speed up the revision process.   

We look forward to receiving your revised manuscript.

Kind regards,

Sebastian D. Fugmann, Ph.D.

Academic Editor

PLOS ONE

Journal Requirements:

Reviewers' comments:

Reviewer's Responses to Questions

**Comments to the Author**

1. Is the manuscript technically sound, and do the data support the conclusions?

Reviewer #1: Yes

Reviewer #2: Yes

Reviewer #3: Yes

2. Has the statistical analysis been performed appropriately and rigorously? 

Reviewer #1: N/A

Reviewer #2: Yes

Reviewer #3: Yes

3. Have the authors made all data underlying the findings in their manuscript fully available?

Reviewer #1: Yes

Reviewer #2: Yes

Reviewer #3: Yes

4. Is the manuscript presented in an intelligible fashion and written in standard English?

Reviewer #1: Yes

Reviewer #2: Yes

Reviewer #3: Yes

5. Review Comments to the Author

Reviewer #1: In the manuscript “Similarity between mutation spectra in hypermutated genomes of rubella virus and SARS-CoV-2 genomes accumulated during the COVID-19 pandemic”, by Klimczak et al., the authors use bioinformatics and statistical tools to compare the mutational spectra of rubella and SARS-CoV-2 single-stranded RNA viruses. They extracted four main conclusions: 1) The mutational spectra are similar, “with C to U as well as A to G and U to C being the most prominent in plus-strand genomic RNA of each virus”; 2) “U to C changes universally showed a preference for loops versus stems in predicted RNA secondary structure”; 3) “C to U changes showed enrichment in the uCn motif, which suggested a subclass of APOBEC cytidine deaminase being a source of these substitutions”; 4) There was an “Enrichment of several other trinucleotide- centered mutation motifs only in SARS-CoV-2 - likely indicative of a mutation process characteristic to this virus.” The analysis is competent, and the presentation suitable for publication. However, as there is an extensive overlap between this manuscript and the recently published work of the group of Dr. Silvo Conticello (Di Giorgio et al., Sci. Adv. 2020; 6: eabb5813), it would be beneficial if the authors identify the similar and distinct conclusions concerning this work. Furthermore, besides the two additional references already cited in the manuscript, a recent paper by Dr. Simmonds (https://doi.org/10.1128/mSphere.00408-20) should also be referenced and discussed, particularly given its thorough analysis of the C to U changes in the Sars-Cov-2 genomes. The authors should engage in a thorough comparison between their data and the recently published results. To make the manuscript shine, it is also perhaps a good idea to highlight what seems to be the only aspect that the four published papers on the mutational spectra of Sars-Cov-2 have failed to notice: conclusion 2 (U to C changes universally show a preference for loops). Below we comment on each of the first three conclusions.

The mutational spectra are similar

Although this reviewer suspects that the contribution of mutations introduced by NGS is negligible, the authors should address this issue in the methods or results, instead of leaving it to the discussion of the G to U changes (Dr. Simmonds deals with this problem compellingly).

The rational beyond the generation of the NoDups, NoDupsFunc, and NoDupsNonFunc is clear. However, by removing the information on the number of repeated mutations the authors could be missing hotspots. It should be possible to use the phylogenetic data to determine whether repeated mutations are due to independent mutational events or a shared founder effect.

“…density values for specific base substitutions cannot be compared directly between two viruses because they were obtained from vastly different genome numbers”. Given that the major focus of the work is on the comparison between the rubeola and Sars-Cov-2 viruses, it is not apparent why the authors did not subsample the Sars-Cov-2 data to equilibrate the genome numbers and perform the comparison. The comparison may not work for lack of power, but that would be another issue.

Figure 2. 1) The definition of the null hypothesis (“positive correlation between rubella and SARS-CoV-2 spectra”) is misleading. Since the P values are all below 0.05, and the authors conclude that there is a correlation, the null hypothesis must be “lack of positive correlation between rubella and SARS-CoV-2 spectra”. 2) There is a slight drop in the correlation when only silent mutations are considered. Could this mean that part of the correlation is explained by selection?

More information is needed on the “ADAR score tool.”

U to C changes universally show a preference for loops

The authors seem to favor the scenario in which A3A mostly drives the U to C changes. Since the group of Conticello favors the action of APOBEC1, it would be useful to read a discussion on the strengths and weaknesses of each scenario. To boost the impact of the manuscript, the authors should also consider changing the title to highlight the possible role of APOBEC3A. Regardless of their decision on the title of the manuscript, it is essential to 1) clarify whether the preference for RNA loops is an exclusive feature of A3A or common to other APOBECs editing RNA and 2) deal with the fact that “none of the trinucleotide motifs containing C to U base substitutions showed loop or stem preference in selection-free SARS-CoV-2 NoDupsNonFunc filtered dataset.” Concerning this latter point, the authors could perhaps estimate how often the significance is lost when considering random subsets of NoDups with the same number of elements of NoDupsNonFunct. In addition, it might be useful to incorporate the number of independent mutations in the same position to increase the sample size and perhaps reach significance in the NoDupsNonFunct. Finally, an analysis of the frequency of codons prone to give rise to non-synonymous changes upon C to U mutation in the stem versus loop regions could help evaluate whether selection explains the bias toward loops. For instance, if the frequency of these codons is considerably higher in the loop regions, than the lack of significance in the NoDupsNonFunct could not be as problematic as it seems.

C to U changes showed enrichment in the uCn motif

It would be helpful to know what is the degree of overlap between the datasets of the studies already published and the data analyzed in this manuscript.

In the introduction, this topic is presented in a very confusing manner. First, the authors focus on the motif preferred by APOBEC3G, which is only known to target (c)DNA. Then, they mention the target preference of APOBEC3A (A3A) and APOBEC3B (A3B) and another trinucleotide that is also a match. It would be better to remove APOBEC3G from the picture and refer the top three (DNA) trinucleotide preferences of A3A and A3B (as percentages). In addition, the authors could use the data available for APOBEC1and A3A to estimate if the trinucleotide motifs in RNA can be predicted from the trinucleotide motifs in DNA (https://doi.org/10.1038/s41467-020-16802-8 provides some hints).

Minor issues

A figure with a model would be useful. Furthermore, the authors could consider a complete map of the loop and stem regions along the Sars-Cov-2 genome. This may require some creativity due to scale issues, but for a quick reading any other form of data presentation is better than a table.

The writing could be improved. There are several passages with weird punctuation or grammar that need editing (e.g., “Similarly, to what,” “with the null hypothesis that, there is,” “genome shown below,” “generating RNA mutation load in population of SARS-CoV-2 genomes”, “in individual isolates We de-duplicated”…)

Reviewer #2: In this manuscript by Dmitry A. Gordenin et al. make effective use of published data sets to address a relevant and straightforward research question: Is there a similarity between mutation spectra in hypermutated genomes of rubella virus and in SARS-CoV-2 genomes accumulated during the COVID-19 pandemic. The genomes of six independent isolates of hypermutated 60 vaccine-derived rubella viruses provide a total of 993 mutations, as previously published. This relatively small data set was compared to a large collection of 32,341 whole genome sequence of multiple SARS-CoV-2 isolates (FASTA entries) were included in the mutation calling, of which 251,273 mutations were retained.

Mutational patterning and motif searches suggest that the mutation mechanisms underlying hypermutation of the rubella vaccine virus are very similar or even identical to those of the SARS-CoV-2 viruses propagating in the human population with regard to the uCn APOBEC signature. An enrichment of several other trinucleotide-centered mutation motifs was also found in the SARS-CoV-2 . Despite the fact, that the latter have not been characterized in further detail and compared (limited rubella data set) this is a relevant and interesting study that deserves publication in its present form.

Reviewer #3: Thank you for the opportunity to review the manuscript of Gordenin et al., on the comparative mutation-spectrum evaluation between two ssRNA viruses (Rubella and SARS-CoV-2), which I had previously read on bioRxiv. The strong points of this study are that the computational methods including the mathematical and statistical calculations that, as applied, make independent and diverse datasets comparable. On the other hand, this study has a few weaknesses with regards to the theoretical background and results explanations that I will summarise point-by-point below. Overall this is the first study that highlights the evolutionary impact of APOBEC and (very likely) ADAR editing of ssRNA viruses from the population-genetics perspective, and also the first one to link this to the mutability and infectivity of the pandemic-responsible SARS-CoV-2. I therefore recommend that the authors add in the discussion a paragraph that walks through a couple of such examples to connect the present study with previous ones. Other than that, this is an excellent study that should be published.

Point-by-point comments

Introduction

• Lines 24-26: RdRPs may be a source of mutation, not a question, however as shown downstream in this work it seems that it’s rather of a “background-frequency” when SNVs are catalogued in aggregate. I suggest adding a “may” be a source of mutation in this sentence.

• Lines 66-71: A U-to-C change on the (+) annotated strand of the ss-RNA viruses rubella and SARS-CoV-2 can only be an ADAR effect only if the complementing (-) strand is not from the same virus-genome, as the authors very correctly highlight in lines 72-74. It can perhaps be another RNA or DNA molecule (intermediate? Another viral genome copy?). When SNVs are called by alignment to single-stranded viral genomes the A-to-G on the “other side” which is formed through a loop will be still read as A-to-G. Which leaves an open question about potentially other modifications that may lead to U-to-C on RNAs. I recommend that the authors should have a look into this too (https://www.nature.com/articles/s41467-019-13525-3) and cite it as it highlights potential other sources for U-to-C changes on the mRNA, though not primarily focusing on that. In addition, the fact that A-to-G and U-to-C are probably not always ADAR-related is also supported by the data presented: in Figure 2 it shows that the A-to-G and T-to-C density values are never the close in the robust datasets (A and B panels). The same misconception is also published in reference [23]. (Lines 356 - 364) additional statistical data support this discrepancy.

Results

• Lines 246-251:This should move at the end of the introduction, between the two sentences ending and starting in line 90. Like this it will set the main goal of this study straight out and help the reader get on board faster.

• Lines 257- 326: This part is very technical I suggest that it gets shortened in the results and be included in the Materials and Methods.

• Lines 253-255: Please, provide a more self explanatory figure legend. Also in 339-346 legend Figure 2 explain the Dups, NoDups etc

• Line 349: For accuracy please consider changing the term “genomic” RNA, which could be very confusing for the reader, and instead I recommend using something like “viral RNA genome”.

• Lines 366-371: there is a good chance here to introduce the contribution of the RdRP errors up to the background level frequency.

• Lines 402-404: This sentence is a bit self-contradicting. Since secondary structure effects can not be predicted, how can the U-to-C can be explained? Maybe these are cases where these changes are not ADAR-driven? It is confusing, please rephrase clearly.

• Lines 424-426: please rephrase here since there’s a separate calculation and visualisation (Fig 4) for C-to-U, G-to-A, U-to-C, A-to-G. Another point that it’s worth keeping in mind is, that in the APOBEC motif sought by the authors it’s not impossible that different enzymes are targeting the (+) or (-) strand in the same double-stranded structure. For instance, the ds structure could be an RNA-DNA hybrid - (some APOBEC3s show preference to such hybrids).

Discussion

• Lines 498-500: this is a very intriguing point which can also be supported by the findings in reference [23] by the no-specificity in the ADAR motif (25% base composition per nucleotide in the flanking bases). Please, make it a separate paragraph and explain how this can work. Moreover, in Lines 500-501: please rephrase and explain in detail what you mean by “ADARs only in vivo”.

Figures

• All figures should be reformatted to a better quality.

• Figure 1: the text should be minimized.

• Figure 2: The panel titles (NoDupsNonFunc etc) should be better explained (also in Figure 3). Perhaps keep the one “dominant” panel (A,B,C?) that really summarizes the mutations in SARS-CoV-2 which can be immediately compared with the corresponding one of Rubella (D). Provide the rest both for SARS-CoV-2 and Rubella in the supplementary material (both for Figure 2 and 3).

• Figure 3: in panels A,B,C there is a higher rate of background density from the least dominant mutations that seems higher than the one in Rubella (D). I presume that this is due to the different depth. However if this is an error rate by the polymerase, please calculate it and perform a statistical comparison (wilcoxon test - to consider the rank association of the background?) to Rubella.

• Figure 4: please remove the 0 to achieve a less busy plot. Again, keep one of the NoDupsNonFunc, NoDups or Dups. Whichever you think is the most representative to be compared to Rubella (the rest put in the supplementary). And please then colour the bars the same way to highlight potential complementarity between C-to-U / G-to-A or A-to-G and U-to-C (which I can already see that there’s no complementarity to A-to-G and U-to-C. So maybe they aren’t really related?) Please discuss this in Lines 498 and 500.

6. PLOS authors have the option to publish the peer review history of their article (what does this mean?). If published, this will include your full peer review and any attached files.

Reviewer #1: No

Reviewer #2: No

Reviewer #3: **Yes: **F Nina Papavasiliou

---

## [Author Response · Author response to Decision Letter 0]

18 Sep 2020

The formatted version of this text is provided as a Word file in the resubmission package and is viewable at the end of combined PDF.

Point by Point Response

to reviewer’s general evaluation and to Comments to the Author

Author’s text in Response is typed in blue font under relevant sections of reviews, which are left in black font. Quotations from the revised text are in red font

All references to text lines are as in the original version.

General evaluation by reviewers.

Reviewer's Responses to Questions

1. Is the manuscript technically sound, and do the data support the conclusions?

Reviewer #1: Yes

Reviewer #2: Yes

Reviewer #3: Yes

2. Has the statistical analysis been performed appropriately and rigorously? 

Reviewer #1: N/A

Reviewer #2: Yes

Reviewer #3: Yes

3. Have the authors made all data underlying the findings in their manuscript fully available?

Reviewer #1: Yes

Reviewer #2: Yes

Reviewer #3: Yes

4. Is the manuscript presented in an intelligible fashion and written in standard English?

Reviewer #1: Yes

Reviewer #2: Yes

Reviewer #3: Yes

Author’s responses to general evaluation :

We thank all three reviewers for their careful analysis of our manuscript and for agreeing that the submitted work is technically sound, all data support the conclusions, and statistical analysis was performed appropriately and rigorously. We also thank for the comments aimed to improve our manuscript. These will be specifically addressed in the point-by-point response below. Preceding the specific responses, it may be helpful to reiterate the main question of this study – is there a similarity between the features of the mutation load slowly accumulating in SARS-CoV-2 genomes during the current pandemic and the burst-like mutation load in the previously discovered hypermutated genomes of attenuated rubella-vaccine virus. This question is important because, while there were already indications of similarity of the SARS-CoV-2 and hypermutated rubella mutation spectra, such similarity has not been validated by rigorous statistical analysis of a large dataset. Our study design and data analysis have provided strong statistical support to such a similarity between several features of the slowly accumulating mutation load in SARS-CoV-2 and the burst-like mutation load accumulated in attenuated rubella-vaccine virus. While this could be a coincidental similarity, it may also be a consequence of similarity between the background mutational processes operating in SARS-CoV-2 and the active mutational processes that resulted in hypermutation of rubella virus. The importance of our question and of the performed analysis is underscored by the fact that the hypermutation of the live attenuated rubella vaccine virus revealed in our previous work generated infectious virus particles. While this was documented only in rare cases, granulomas of immunocompromised children, the knowledge about potential background level presence of similar mutational mechanisms in SARS-CoV-2 should be accounted in future studies.

Along with the reviewers, we state that our data support this conclusion as well as the conclusions about several incidental findings. Therefore, while the additional analyses suggested below may lead to more incidental findings, they are not required to support the conclusions of our work. Moreover, the suggested analyses appear challenging or technically difficult and will just delay publication of our conclusions that have already been validated by all three reviewers. We have addressed the points raised by the suggested analyses in the point-by-point response below.

We also thank the reviewers and Editor for the specific comments aimed to improve our manuscript. In response to reviewer’s and Editor’s suggestions we extended citation of studies that came to conclusions similar to ours in the part related specifically to SARS-CoV-2. To the best of our knowledge, no detailed comparison between SARS-CoV-2 and hypermutated rubella mutation spectra and signatures have been published so far. These as well as all other reviewer’s comments will be addressed individually in the point-by-point response below.

All references to text lines are as in the original version.

Author’s text in the response is shown in blue. Changes in the text quoted in the response are shown in red both here and in a separately uploaded marked-up copy.

Point-by-point-response to:

5. Review Comments to the Author

Reviewer #1:

In the manuscript “Similarity between mutation spectra in hypermutated genomes of rubella virus and SARS-CoV-2 genomes accumulated during the COVID-19 pandemic”, by Klimczak et al., the authors use bioinformatics and statistical tools to compare the mutational spectra of rubella and SARS-CoV-2 single-stranded RNA viruses. They extracted four main conclusions: 1) The mutational spectra are similar, “with C to U as well as A to G and U to C being the most prominent in plus-strand genomic RNA of each virus”; 2) “U to C changes universally showed a preference for loops versus stems in predicted RNA secondary structure”; 3) “C to U changes showed enrichment in the uCn motif, which suggested a subclass of APOBEC cytidine deaminase being a source of these substitutions”; 4) There was an “Enrichment of several other trinucleotide- centered mutation motifs only in SARS-CoV-2 - likely indicative of a mutation process characteristic to this virus.” The analysis is competent, and the presentation suitable for publication.

We thank the reviewer for the accurate summary and for the positive evaluation of our original submission.

However, as there is an extensive overlap between this manuscript and the recently published work of the group of Dr. Silvo Conticello (Di Giorgio et al., Sci. Adv. 2020; 6: eabb5813), it would be beneficial if the authors identify the similar and distinct conclusions concerning this work.

Di Giorgio et al., 2020 was cited (ref. 23 in the initial submission, ref. 26 in revision) in two places of our manuscript as well as two other publications (ref 32, 33 in the initial version, refs 36, 37 in revision) suggesting the roles for RNA editing enzymes in generating the mutation load in SARS-CoV-2. 

Furthermore, besides the two additional references already cited in the manuscript, a recent paper by Dr. Simmonds (https://doi.org/10.1128/mSphere.00408-20) should also be referenced and discussed, particularly given its thorough analysis of the C to U changes in the Sars-Cov-2 genomes. 

We have added a reference to Simmonds, 2020 (new ref 27 on lines 86 and 377). This and the other above mentioned citations are associated with the previously existing text (lines 83-86, lines 375-379) and with the modified text (lines 483-486 shown below) pointing to similarity with our analysis in the part of high abundance of C to U mutations as well as A to G mutations.

Modification in lines 483-486:

In agreement with the previous analyses performed separately for rubella [20] and for SARS-CoV2 [26, 27, 36, 37] comparisons between SARS-CoV-2 and hypermutated rubella strains demonstrated that the base substitution spectra correlate between the two viruses. Two types of base substitutions – C to U (and its complementary G to A) and A to G ( and its complementary U to C), expected from endogenous mutagenesis by APOBECs and ADARs, respectively, prevailed in both viruses (Fig 2). [a portion of the old text also shown in black to provide the logical context].

The authors should engage in a thorough comparison between their data and the recently published results. 

Even if the suggested comparisons were feasible, the additional analysis would not alter any conclusions of our manuscript. Since the data and the analyses formats in those prior works are different from ours, the analytical comparisons are not likely to be possible or would be at least technically difficult and would just delay publication of our conclusions that have already been validated by all three reviewers. We also note that the comparison of the fine details of SARS-CoV-2 mutation spectra revealed by our statistical analyses with the other SARS-CoV-2 studies goes beyond the scope of our study devoted to comparison of SARS-CoV-2 and rubella mutation loads.

To make the manuscript shine, it is also perhaps a good idea to highlight what seems to be the only aspect that the four published papers on the mutational spectra of Sars-Cov-2 have failed to notice: conclusion 2 (U to C [reviewer probably meant C to U] changes universally show a preference for loops). 

This feature was sufficiently highlighted by a separate section of Results (lines 373-408) and in Discussion (lines 488-493 and lines 503-507), Figure 3, S3 Fig., section of Table 1. Since this is an incidental finding that would benefit from validation in more data analyses and experimental studies, we do not see the need for additional highlighting of this finding.

Below we comment on each of the first three conclusions.

The mutational spectra are similar

Although this reviewer suspects that the contribution of mutations introduced by NGS is negligible, the authors should address this issue in the methods or results, instead of leaving it to the discussion of the G to U changes (Dr. Simmonds deals with this problem compellingly).

In line with the scope of our study to compare the mutation spectra of SARS-CoV-2 and hypermutated rubella, we concentrated on G to U changes because these changes showed the only detectable discrepancy between the two viruses (lines 366-371). In the discussion on lines 519-528 we reasoned (in agreement with this reviewer) that sequencing errors are unlikely to be the cause of such a discrepancy.

We thank the reviewer for pointing to the analysis of Dr. Simmons indicating that a low chance of sequencing errors significantly shifts the SARS-CoV-2 mutation spectrum in the dataset and we have inserted the following passage on line 527:

We also note that the recent study [27] indicated the overall low chance of sequencing errors reflected in SARS-CoV-2 consensus sequences in the dataset analyzed in that work.

The rational beyond the generation of the NoDups, NoDupsFunc, and NoDupsNonFunc is clear. However, by removing the information on the number of repeated mutations the authors could be missing hotspots. It should be possible to use the phylogenetic data to determine whether repeated mutations are due to independent mutational events or a shared founder effect.

Looking for hotspots in the SARS-CoV-2 mutation load and distinguishing between functional selection versus mutagenic mechanisms preference is beyond the scope of our study and requires extensive separate analyses exploring several methods and statistical hypotheses. Such analyses can be certainly attempted by others using the complete list of 251,273 unfiltered redundant mutation calls generated by our study and provided in S1A Table.

“…density values for specific base substitutions cannot be compared directly between two viruses because they were obtained from vastly different genome numbers”. Given that the major focus of the work is on the comparison between the rubeola and Sars-Cov-2 viruses, it is not apparent why the authors did not subsample the Sars-Cov-2 data to equilibrate the genome numbers and perform the comparison. The comparison may not work for lack of power, but that would be another issue.

We agree with the reviewer that such a comparison would lack power because most of the viral RNA genomes would contain the majority or even all mutations coming from the predecessors present in the same dataset. Determining the complete set of new mutations in each viral RNA genome would require distinguishing between the roles of functional selection and mutational preferences which cannot be done unambiguously. We noted that unlike the SARS-CoV-2 dataset, each mutation in each hypermutated rubella virus occurred independently from similar mutations in other rubella RNA genomes present in the same dataset (see lines 304-306 of the original submission).

Figure 2. 

1) The definition of the null hypothesis (“positive correlation between rubella and SARS-CoV-2 spectra”) is misleading. Since the P values are all below 0.05, and the authors conclude that there is a correlation, the null hypothesis must be “lack of positive correlation between rubella and SARS-CoV-2 spectra”. 

Thank you for noticing this inaccurate statement. The requested change has been entered into the legend of Figure 2 (line 345).

2) There is a slight drop in the correlation when only silent mutations are considered. Could this mean that part of the correlation is explained by selection?

There could be several explanations for this really “slight drop”, in particular the lower statistical power provided by this smallest filtered subset containing only 4,740 mutation events as compared with the other two larger subsets (7,416 and 12,156 mutations) analyzed (see numbers in Figure 1) 

More information is needed on the “ADAR score tool.”

We have added the following to S1 Fig legend (line 760):

ADAR scores were calculated using the Web tool http://hci-bio-app.hci.utah.edu:8081/Bass/InosinePredict

U to C changes universally show a preference for loops

The authors seem to favor the scenario in which A3A mostly drives the U to C changes. Since the group of Conticello favors the action of APOBEC1, it would be useful to read a discussion on the strengths and weaknesses of each scenario.

We do not suggest that APOBEC3A is the most likely candidate for making C to U mutations in SARS-CoV-2 genome. We mention only APOBEC3A, but not the other APOBECs, in discussion of loop-over-stem preference, because so far it is the only APOBEC enzyme with a documented biochemical preference for stems over loops, however this feature may be also among the characteristics of other APOBECs, including APOBEC1 acting on viral RNA. Model studies of APOBEC editing of viral RNAs, including the models with expression of individual APOBECs, is required for justified speculations on this subject, which we already stressed on lines 62-68 in the original submission. In order to eliminate the impression that APOBEC3A is the favorite hypothesis we have added the following passage on line 507:

…which [APOBEC3A] so far is the only RNA-editing APOBEC explored for this feature.

To boost the impact of the manuscript, the authors should also consider changing the title to highlight the possible role of APOBEC3A. 

Based on the above considerations, we feel it would be premature to include APOBEC3A in the title.

Regardless of their decision on the title of the manuscript, it is essential to 

1) clarify whether the preference for RNA loops is an exclusive feature of A3A or common to other APOBECs editing RNA 

See our addition to line 507 described above.

and 2) deal with the fact that “none of the trinucleotide motifs containing C to U base substitutions showed loop or stem preference in selection-free SARS-CoV-2 NoDupsNonFunc filtered dataset.” Concerning this latter point, the authors could perhaps estimate how often the significance is lost when considering random subsets of NoDups with the same number of elements of NoDupsNonFunct.

The suggested analysis will not affect our conclusions (which this reviewer agreed to be supported by the data) regardless of the results. If “significance is lost” with some subsets it could be caused by the lower statistical power provided by the smaller sample sizes.

In addition, it might be useful to incorporate the number of independent mutations in the same position to increase the sample size and perhaps reach significance in the NoDupsNonFunct.

Different substitutions of the same nucleotide were already included in NoDupsNonFunct, wherever such substitutions occurred in the same nucleotide (see lines 126-128).

Finally, an analysis of the frequency of codons prone to give rise to non-synonymous changes upon C to U mutation in the stem versus loop regions could help evaluate whether selection explains the bias toward loops. For instance, if the frequency of these codons is considerably higher in the loop regions, than the lack of significance in the NoDupsNonFunct could not be as problematic as it seems.

Both, mutations in non-coding regions as well as mutations in coding regions resulting in synonymous codons were included in Non-functional category so the results of the suggested analysis would allow more than a single interpretation.

C to U changes showed enrichment in the uCn motif

It would be helpful to know what is the degree of overlap between the datasets of the studies already published and the data analyzed in this manuscript.

GSAID was the most complete dataset of SARS-CoV-2 RNA genome sequences and included most of the pandemic related SARS-CoV-2 genomes analyzed in prior publications. Specific overlaps can be determined by comparing the Excel acknowledgement tables that list the origins of SARS-CoV-2 sequences in each publication. Relative re-evaluation of the statistical power defining the specific features of SARS-CoV-2 mutation spectra in the different publications is beyond the scope of our study aimed at comparison of the mutation loads of SARS-CoV-2 and hypermutated rubella viruses. This said, we have highlighted other works that came to similar conclusions about the prevalence of C to U (G to A) and A to G (U to C) changes in SARS-CoV-2 (discussed above and referenced in the revision as 25, 26, 35, 36).

In the introduction, this topic [“this” apparently refers to “C to U changes showed enrichment in the uCn motif”] is presented in a very confusing manner. First, the authors focus on the motif preferred by APOBEC3G, which is only known to target (c)DNA. Then, they mention the target preference of APOBEC3A (A3A) and APOBEC3B (A3B) and another trinucleotide that is also a match. It would be better to remove APOBEC3G from the picture and refer the top three (DNA) trinucleotide preferences of A3A and A3B (as percentages). 

In fact, the confusion about this section of Introduction was caused by our accidentally dropping a reference implying RNA editing capability for APOBEC3G. We have brought back the reference to A3G RNA editing [new ref 13] to line 46.

In addition, the authors could use the data available for APOBEC1and A3A to estimate if the trinucleotide motifs in RNA can be predicted from the trinucleotide motifs in DNA (https://doi.org/10.1038/s41467-020-16802-8 provides some hints).

The suggested paper by Jalili et al., 2020 (already referenced (ref 35 in the original submission; ref 39 in revision) does not contain any mention of APOBEC1

Minor issues

A figure with a model would be useful. Furthermore, the authors could consider a complete map of the loop and stem regions along the Sars-Cov-2 genome. This may require some creativity due to scale issues, but for a quick reading any other form of data presentation is better than a table.

Models of plus-strand ssRNA virus replication and potential mutation generation are provided in several reviews cited in our work as well as in Conticello group recent paper (ref 23 in the original submission, ref 26 in revision). We believe that the summary of preferences organized in Table 1 better serves the purpose of discussion targeted to the main questions addressed by our analysis.

The writing could be improved. There are several passages with weird punctuation or grammar that need editing (e.g., “Similarly, to what,” “with the null hypothesis that, there is,” “genome shown below,” “generating RNA mutation load in population of SARS-CoV-2 genomes”, “in individual isolates We de-duplicated”…)

We have corrected punctuation and grammar in the positions quoted above (lines 12, 152, 201, 247, 283).

In summary, we reiterate our thanks to this reviewer for the careful analysis and positive evaluation of our original submission. We hope to use some of their suggestions in future analyses of RNA editing signatures that go beyond the scope of this submission.

Reviewer #2: 

In this manuscript by Dmitry A. Gordenin et al. make effective use of published data sets to address a relevant and straightforward research question: Is there a similarity between mutation spectra in hypermutated genomes of rubella virus and in SARS-CoV-2 genomes accumulated during the COVID-19 pandemic. The genomes of six independent isolates of hypermutated 60 vaccine-derived rubella viruses provide a total of 993 mutations, as previously published. This relatively small data set was compared to a large collection of 32,341 whole genome sequence of multiple SARS-CoV-2 isolates (FASTA entries) were included in the mutation calling, of which 251,273 mutations were retained.

Mutational patterning and motif searches suggest that the mutation mechanisms underlying hypermutation of the rubella vaccine virus are very similar or even identical to those of the SARS-CoV-2 viruses propagating in the human population with regard to the uCn APOBEC signature. An enrichment of several other trinucleotide-centered mutation motifs was also found in the SARS-CoV-2 . Despite the fact, that the latter have not been characterized in further detail and compared (limited rubella data set) this is a relevant and interesting study that deserves publication in its present form.

We thank this reviewer for the positive evaluation of our work. Several differences between the rubella and SARS-CoV-2 mutational spectra and signatures were highlighted in Discussion. In the last paragraph of Discussion, we speculated on the future diagnostic use of one of these differences.

Reviewer #3:

Thank you for the opportunity to review the manuscript of Gordenin et al., on the comparative mutation-spectrum evaluation between two ssRNA viruses (Rubella and SARS-CoV-2), which I had previously read on bioRxiv. The strong points of this study are that the computational methods including the mathematical and statistical calculations that, as applied, make independent and diverse datasets comparable.

We thank the reviewer for highlighting the main reason that made us to undertake this analysis derived from the statistical methods of evaluating tri- (or tetra-)nucleotide-centered mutational signatures that we had initially developed for analyzing somatic mutagenesis in human genome (refs. 14 and 26 in the original submission, refs 15 and 30 in revision).

On the other hand, this study has a few weaknesses with regards to the theoretical background and results explanations that I will summarise point-by-point below.

We have addressed several apparent weakness points in the point-by-point response below.

Overall this is the first study that highlights the evolutionary impact of APOBEC and (very likely) ADAR editing of ssRNA viruses from the population-genetics perspective, and also the first one to link this to the mutability and infectivity of the pandemic-responsible SARS-CoV-2. I therefore recommend that the authors add in the discussion a paragraph that walks through a couple of such examples to connect the present study with previous ones. Other than that, this is an excellent study that should be published.

A note about the comparison with previous studies was inserted at the beginning of the second paragraph of Discussion (see also response to reviewer 1). Along with a general PLOS One policy we did not stressed on the novelty of conclusions following from our analyses, but just indicated related resuls of other groups.

Point-by-point comments

Introduction

• Lines 24-26: RdRPs may be a source of mutation, not a question, however as shown downstream in this work it seems that it’s rather of a “background-frequency” when SNVs are catalogued in aggregate. I suggest adding a “may” be a source of mutation in this sentence.

Done

• Lines 66-71: A U-to-C change on the (+) annotated strand of the ss-RNA viruses rubella and SARS-CoV-2 can only be an ADAR effect only if the complementing (-) strand is not from the same virus-genome, as the authors very correctly highlight in lines 72-74. It can perhaps be another RNA or DNA molecule (intermediate? Another viral genome copy?). When SNVs are called by alignment to single-stranded viral genomes the A-to-G on the “other side” which is formed through a loop will be still read as A-to-G. Which leaves an open question about potentially other modifications that may lead to U-to-C on RNAs. I recommend that the authors should have a look into this too (https://www.nature.com/articles/s41467-019-13525-3) and cite it as it highlights potential other sources for U-to-C changes on the mRNA, though not primarily focusing on that. In addition, the fact that A-to-G and U-to-C are probably not always ADAR-related is also supported by the data presented: in Figure 2 it shows that the A-to-G and T-to-C density values are never the close in the robust datasets (A and B panels). The same misconception is also published in reference [23]. (Lines 356 - 364) additional statistical data support this discrepancy.

A possibility that A to G and U to C changes in SARS-CoV-2 may not be stemming from ADAR action was already presented in the original submission (lines 499-500). We thank the reviewer for highlighting uracil modifications as a possible other source for U to C (or A to G) changes. We have added the following text:

at the end of line 76:

Besides ADAR editing, U to C (or complementary A to G) changes can result from uracil modifications by enzymes normally acting on specific uracils in tRNAs [23, 24].

Results

• Lines 246-251:This should move at the end of the introduction, between the two sentences ending and starting in line 90. Like this it will set the main goal of this study straight out and help the reader get on board faster.

Thank you for this good suggestion. We have moved this passage as suggested.

• Lines 257- 326: This part is very technical I suggest that it gets shortened in the results and be included in the Materials and Methods.

Should these lines be moved to Material and Methods, they would have to be split between several parts of this section. We preferred to leave it in one place preceding the description of the outputs, because it defines the structure of the filtered data used in the analyses.

• Lines 253-255: Please, provide a more self explanatory figure legend. 

A self-explanatory Figure 1 legend (lines 253-255) would require repeating a significant part of the section text. Instead, we have added the following clarifying sentence to the Figure 1 legend:

Details of mutation call filtering and grouping as well as abbreviations are explained in the text of the "Design of the analysis" section.

Also in 339-346 legend Figure 2 explain the Dups, NoDups etc

We have added the following clarification at the beginning of the Figure 2, Figure 3 and Figure 4 legends

Creation of filtered SARS-CoV-2 MAFs and their abbreviated names NoDupsNonFunc, NoDupsFunc, NoDups, are described in Fig 1 and in associated text.

• Line 349: For accuracy please consider changing the term “genomic” RNA, which could be very confusing for the reader, and instead I recommend using something like “viral RNA genome”.

Replaced as suggested

• Lines 366-371: there is a good chance here to introduce the contribution of the RdRP errors up to the background level frequency.

Lines 366-371 of the result section introduced G to U changes which are prominent in SARS-CoV-2, but are nearly absent in rubella. In lines 521-528 of the original submission, we did discuss one of the several possible origins of these discrepancies between the SARS-CoV-2 and rubella spectra. While we cannot exclude an RdRP error as a potential source of such a discrepancy, we felt it would be premature to invoke this speculation because Coronaviridae have a mechanism of RdRP error correction, while rubella does not.

• Lines 402-404: This sentence is a bit self-contradicting. Since secondary structure effects can not be predicted, how can the U-to-C can be explained? Maybe these are cases where these changes are not ADAR-driven? It is confusing, please rephrase clearly.

Rephrased lines 402-404 as follows:

If the U to C (A to G) changes were to come from ADAR adenine deaminase acting on dsRNA, secondary structure effects of ssRNA intermediate folding would not be expected. Alternatively, these changes could be not ADAR driven.

• Lines 424-426: please rephrase here since there’s a separate calculation and visualisation (Fig 4) for C-to-U, G-to-A, U-to-C, A-to-G. 

Rephrased as follows:

Therefore, we concentrated on the motifs representing the most abundant types of base substitutions present in both viruses, i.e., on the C to U and their complement G to A, as well as U to C and their complement A to G changes.

Another point that it’s worth keeping in mind is, that in the APOBEC motif sought by the authors it’s not impossible that different enzymes are targeting the (+) or (-) strand in the same double-stranded structure. For instance, the ds structure could be an RNA-DNA hybrid - (some APOBEC3s show preference to such hybrids).

By our knowledge, DNA-RNA hybrids have not been reported in the replication cycles of rubella and SARS-CoV-2

Discussion

• Lines 498-500: this is a very intriguing point which can also be supported by the findings in reference [23] by the no-specificity in the ADAR motif (25% base composition per nucleotide in the flanking bases). Please, make it a separate paragraph and explain how this can work. 

The ADAR scores as well as the online tool for ADAR score calculation was cited in the original submission as ref 8 and were described on lines 34-41. We have added URL of the web tool to the legend of S1 Fig (see response to reviewer 1). We note that the result of the ADAR score analysis was negative and thus could have multiple explanations, including insufficient sample size. Thus, we did not further elaborate on details of the tool, which are well described in ref 8 and at the Web location.

Moreover, in Lines 500-501: please rephrase and explain in detail what you mean by “ADARs only in vivo”.

Replaced as the text in lines 500-501 to account for this, as well as for the below comment to lines 498-500 as follows:

If A to G or U to C changes are really stemming from ADAR activity, it is possible that these enriched motifs are preferred by ADARs in SARS-CoV-2 but not in RNA substrates used to generate the editing consensus in [8]. Alternatively, A to G and U to C changes could be caused by one or more mechanisms unconnected to ADARs.

Figures

• All figures should be reformatted to a better quality.

All main and supplementary figures used for the original submission were verified with PACE web tool. High resolution main figures were available via links in the combined PDF created by the PLOS One submission system

• Figure 1: the text should be minimized.

We respectfully disagree. We feel that Figure 1, as presented in the original submission, allowed all three reviewers to grasp the details of our analysis rationale and workflow.

• Figure 2: The panel titles (NoDupsNonFunc etc) should be better explained (also in Figure 3). Perhaps keep the one “dominant” panel (A,B,C?) that really summarizes the mutations in SARS-CoV-2 which can be immediately compared with the corresponding one of Rubella (D). Provide the rest both for SARS-CoV-2 and Rubella in the supplementary material (both for Figure 2 and 3).

Panel title details were added to Figures 2, 3, 4 legends. The reasons for comparing the rubella data with each of the three filtered SARS-CoV-2 MAFs (NoDupsNonFunc, NoDupsFunc, NoDups) was explained on lines 301-315. In short, selection would be a confounding factor in each of these comparisons by an extent that is not possible to evaluate. Thus, in our view, presenting all three comparisons on main figures would add convenience, because it would not require a reader continuously switching between the main and supplemental display items.

• Figure 3: in panels A,B,C there is a higher rate of background density from the least dominant mutations that seems higher than the one in Rubella (D). I presume that this is due to the different depth. However if this is an error rate by the polymerase, please calculate it and perform a statistical comparison (wilcoxon test - to consider the rank association of the background?) to Rubella.

Lines 330-337 in the original submission explain the reason, i.e. the vastly different number of SARS-CoV-2 genomes vs the number of rubella genomes used to generate the mutation catalogs, by which we cannot directly compare densities of base substitutions between SARS-CoV2 and rubella and therefore should only compare distribution of base substitution densities obtained from each dataset.

• Figure 4: please remove the 0 to achieve a less busy plot. 

Again, keep one of the NoDupsNonFunc, NoDups or Dups. Whichever you think is the most representative to be compared to Rubella (the rest put in the supplementary). And please then colour the bars the same way to highlight potential complementarity between C-to-U / G-to-A or A-to-G and U-to-C (which I can already see that there’s no complementarity to A-to-G and U-to-C. 

We understand that Figure 4 is busy and may appear counter-intuitive, because, unlike in mutagenesis of genomic DNA, viral strands can mutate independently. Therefore showing complementary nucleotides by the same color may unintentionally mislead a reader. Instead (as was already explained in Figure 4 legend) we annotated complementary motifs under each graph and highlighted in red the few cases in which statistically significant enrichment was detected in complementary motifs. These cases are summarized in Table 1 and in Discussion. The reason for not splitting the results obtained with the three filtered MAFs (NoDupsNonFunc, NoDupsFunc and NoDups) is provided in the above response to a comment about Figure 2. This said, we agree that all efforts should be made to help the reader going through this figure. Therefore we have supplemented the Figure 4 legend with the following text:

Creation of filtered SARS-CoV-2 MAFs and their abbreviated names NoDupsNonFunc, NoDupsFunc, NoDups are described in Fig 1 and in the associated text. The order of data in each group is the same as in the panel’s legend - Rubella, NoDupsNonFunc, NoDupsFunc, NoDups. Zero values are shown for the convenience of following the order of values within each group.

So maybe they aren’t really related?) Please discuss this in Lines 498 and 500.

We have added the following text at the end of this paragraph:

Alternatively, A to G and U to C changes could be caused by one or more mechanisms unconnected to ADARs.

In summary, we thank this reviewer for the high evaluation of our manuscript, careful analysis and useful advice that we have tried to follow as much as possible.

Additional changes advised by an expert in rubella virus:

A. Replaced “rubella” with “rubella virus” in several places, where the word “rubella” may be misinterpreted as referring to a disease rather that to a virus.

B. Clarified the nature of rubella vaccine virus to stress biological effect of hypermutation.

Lines 472-473:

Previously, we demonstrated that hypermutation of the live attenuated rubella vaccine virus [42] can generate infectious virus particles in immunocompromised children [20].

[Shown in context with pre-existing text to stress the rationale for the change]

---

## [Editor Report · Decision Letter 1]

22 Sep 2020

Similarity between mutation spectra in hypermutated genomes of rubella virus and in SARS-CoV-2 genomes accumulated during the COVID-19 pandemic

PONE-D-20-23843R1

Dear Dr. Gordenin,

We’re pleased to inform you that your manuscript has been judged scientifically suitable for publication and will be formally accepted for publication once it meets all outstanding technical requirements.

Kind regards,

Sebastian D. Fugmann, Ph.D.

Academic Editor

PLOS ONE
---

## [Editor Report · Acceptance letter]

24 Sep 2020

PONE-D-20-23843R1 

Similarity between mutation spectra in hypermutated genomes of rubella virus and in SARS-CoV-2 genomes accumulated during the COVID-19 pandemic 

Dear Dr. Gordenin:

I'm pleased to inform you that your manuscript has been deemed suitable for publication in PLOS ONE. Congratulations! Your manuscript is now with our production department. 

Kind regards, 

on behalf of

Dr. Sebastian D. Fugmann 

Academic Editor

PLOS ONE